# *HDAC1*-Mediated lncRNA Stimulatory Factor of Follicular Development to Inhibit the Apoptosis of Granulosa Cells and Regulate Sexual Maturity through miR-202-3p-*COX1* Axis

**DOI:** 10.3390/cells12232734

**Published:** 2023-11-29

**Authors:** Xiaofeng Zhou, Yingting He, Hongyan Quan, Xiangchun Pan, Yinqi Zhou, Zhe Zhang, Xiaolong Yuan, Jiaqi Li

**Affiliations:** Guangdong Laboratory of Lingnan Modern Agriculture, National Engineering Research Center for Breeding Swine Industry, State Key Laboratory of Swine and Poultry Breeding Industry, Guangdong Provincial Key Laboratory of Agro-Animal Genomics and Molecular Breeding, College of Animal Science, South China Agricultural University, Guangzhou 510642, China; zxf93715@163.com (X.Z.); 15521059247@163.com (Y.H.); quanhongyan2022@163.com (H.Q.); 15186255977@126.com (X.P.); zhouyinqi997@163.com (Y.Z.); zhezhang@scau.edu.cn (Z.Z.)

**Keywords:** sexual maturity, follicular development, granulosa cells, histone acetylation, lncRNAs

## Abstract

Abnormal sexual maturity exhibits significant detrimental effects on adult health outcomes, and previous studies have indicated that targeting histone acetylation might serve as a potential therapeutic approach to regulate sexual maturity. However, the mechanisms that account for it remain to be further elucidated. Using the mouse model, we showed that Trichostatin A (TSA), a histone deacetylase (HDAC) inhibitor, downregulated the protein level of Hdac1 in ovaries to promote the apoptosis of granulosa cells (GCs), and thus arrested follicular development and delayed sexual maturity. Using porcine GCs as a cell model, a novel sexual maturity-associated lncRNA, which was named as the stimulatory factor of follicular development (*SFFD*), transcribed from mitochondrion and mediated by *HDAC1*, was identified using RNA sequencing. Mechanistically, *HDAC1* knockdown significantly reduced the H3K27ac level at the −953/−661 region of *SFFD* to epigenetically inhibit its transcription. *SFFD* knockdown released miR-202-3p to reduce the expression of cyclooxygenase 1 (*COX1*), an essential rate-limited enzyme involved in prostaglandin synthesis. This reduction inhibited the proliferation and secretion of 17β-estradiol (E2) while promoting the apoptosis of GCs. Consequently, follicular development was arrested and sexual maturity was delayed. Taken together, *HDAC1* knockdown-mediated *SFFD* downregulation promoted the apoptosis of GCs through the miR-202-3p-*COX1* axis and lead to delayed sexual maturity. Our findings reveal a novel regulatory network modulated by *HDAC1*, and *HDAC1*-mediated *SFFD* may be a promising new therapeutic target to treat delayed sexual maturity.

## 1. Introduction

Sexual maturity is a complex transitional phase in which adolescents acquire sexual maturation and reproductive capacity [1], and its early or late initiation seriously affects adolescent’s mental and physical illnesses, such as depression [2] and polycystic ovarian syndrome (PCOS) [3]. Follicular maturation and the secretion of steroid hormones closely regulate sexual maturity [4,5]. An increase in follicular maturation is associated with precocious sexual maturity, while a decrease in follicular maturation is associated with delayed sexual maturity in adolescent girls [5]. Previous studies have reported that delayed sexual maturity is often marked with arrested follicular development [6,7], reduced maturation rate and quality of oocytes [8,9,10], and is also the earliest manifestation of PCOS [3,11]. However, the molecular therapeutic targets regulating sexual maturity remain unclear.

Histone acetylation, the most extensively studied and appreciated modification, exhibits an essential role in modulating chromatin structure to regulate gene transcription [12]. Previous studies have demonstrated that the disorders of DNA methylation and histone acetylation contribute to delayed sexual maturity [13,14]. Tomikawa et al. reported that 17β-estradiol (E2) increases the expression of *Kiss1*, a sexual maturity-activating gene, through elevating the level of histone H3 acetylation. Notably, the acetylation level in the *Kiss1* promoter is remarkably higher in proestrus animals compared to diestrus animals [15]. Decreased levels of H3K27ac at the *IGF1* promoter in fetal ovarian granulosa cells (GCs) notably inhibit the synthesis of E2 and follicular development [16]. Downregulation of histone deacetylase 3 (*HDAC3*) in GCs elevates the level of H3K14ac and promotes the binding of SP1 in the *Areg* promoter to initiate transcription and follicular maturation [17]. In GCs, GDF9 and BMP15 collaboratively recruit p300 to the *AMH* promoter, leading to elevated histone H3K27ac and markedly increased the expression of *AMH*, a natural gatekeeper of follicle growth and female reproduction [18]. These observations indicate that targeting histone acetylation may serve as a potential therapeutic strategy to regulate sexual maturity in mammals.

The growth and function of GCs exhibit a dominant role in follicular maturation and sexual maturity [19,20]. The proliferation and E2 secretion of GCs facilitate follicular growth [21,22], while the excessive apoptosis of GCs results in massive follicular atresia [23]. Moreover, our previous study indicated that the growth of GCs participates in follicular maturation and sexual maturity [19]. These findings reveal that histone acetylation may regulate the function of GCs to control sexual maturity.

To characterize the mechanisms for histone acetylation controlling sexual maturity, Trichostatin A (TSA), an HDAC inhibitor, was treated with mice to explore the effects of histone acetylation on follicular development and sexual maturity. The physiological mechanisms of porcine follicular development are more similar to those of humans than mice [24]. Therefore, in our study, porcine GCs were used as a cellular model in vitro. Here, we found a novel sexual maturity-associated lncRNA, which was named as the stimulatory factor of follicular development (*SFFD*). Next, we aimed to explore the molecular mechanisms which histone acetylation regulated *SFFD* transcription and how *SFFD* regulated sexual maturation by affecting the functions of GCs. The regulatory mechanism and effects of the *SFFD*-miR-202-3p-*COX1* axis on the proliferation, apoptosis, and estrogen secretion of GCs, follicular development, and sexual maturation were verified. Our results provide a new piece of evidence for *HDAC1*-mediated *SFFD* in the treatment of delayed sexual maturity.

## 2. Materials and Methods

### 2.1. Samples and Chemical Treatment

Three-week-old C57BL/6J mice (36 female and 4 male) from the Guangdong medical laboratory animal center were randomly assigned to three groups: blank (*n* = 12), DMSO (*n* = 12), and TSA (*n* = 12) group. Mice were treated with TSA (HY-15144, MedChemExpress, purity: 99.53%, 0.5 mg/kg) and an equal volume of DMSO for three weeks. Mice were injected 3 times weekly. Among them, 8 mice in each group were randomly selected for counting the age of sexual maturity (known as the pubertal initiation and indicated by the vaginal opening [25]), and their ovaries were used for HE staining, dUTP Nick-End Labeling (TUNEL), immunofluorescence, and Western blot. Two independent researchers examined the vaginal opening of mice for the visual appearance of an opening [26]. The number of follicles in each group was counted in one ovary section (the largest cross-section of each mouse ovary) with three biological replicates. The female mice in each group were mated with the 4 male mice to measure fertility (litter birth weight and number of little size).

### 2.2. Culture of GCs and Real-Time Quantitative PCR (qPCR)

Fresh pig ovaries were collected from a slaughterhouse and maintained in phosphate-buffered saline (PBS) containing penicillin and streptomycin during transporting. Porcine GCs were extracted from the follicular fluid of 3–5 mm follicles with a syringe and washed with PBS. Then, the GCs were cultured in DMEM containing 10% fetal bovine serum and finally incubated at 37 °C under 5% CO_2_. GCs were transfected with vectors using Lipofectamine^TM^ 3000 Reagent (Thermo, Waltham, MA, USA). In total, 100 ng/mL GDF9 (MCE, USA) and 50 ng/mL BMP15 (MCE, USA) were used to treat GCs for 24 h.

The purified total RNA was used as a template for cDNA synthesis with the RevertAid First Strand cDNA Synthesis Kit (Thermo, USA). qPCR was conducted using the SYBR Green qRT-PCR Master with the Bio-Rad CFX96 Touch Real-Time PCR system. The procedure steps were as follows: 95 °C for 10 min, 40 cycles at 95 °C for 15 s, annealing at 60 °C for 30 s. The levels of genes were calculated with the 2^−ΔΔct^ method, and the primer sequences used in pig and mouse are listed in Table 1 and Table 2, respectively.

### 2.3. RNA Sequencing (RNA-seq)

The process and details of RNA-seq were reported in our previous study [27]. Differential screening was conducted using DESeq2 with |log2FC| > 1 and FDR < 0.05.

### 2.4. EdU Assay

An EdU assay was conducted to detect the proliferation of GCs using the Cell-Light^TM^ Edu Kit (RiboBio, Guangdong, Guangzhou, China). Briefly, GCs were treated with 50 µM EdU at room temperature for 2 h, 80% acetone for 30 min, 0.5% Triton X-100 for 10 min, 1 × Apollo for 30 min, and Hoechst for 30 min.

### 2.5. Flow Cytometry and Caspase 3/7 Activity Assay

Flow cytometry was conducted to detect the apoptosis of GCs using the Annexin V-FITC Apoptosis Detection Kit (BioVision, Milpitas, CA, USA). Briefly, GCs were incubated with Annexin V-FITC and propidium iodide for 15 min. The Caspase 3/7 Activity Assay was also performed to detect the apoptosis of GCs using the Caspase 3/7 Activity Assay Kit (Elabscience, Wuhan, China). A total of 50 μL of GCs lysis in 96-well plates was added along with 45 μL 2xReaction buffer and 5 μL of Ac-DEVD-pNA for 2 h. An absorbance of 405 nm was detected.

### 2.6. ELISA

The mouse E2 (ml063198, Enzyme-linked Biotechnology, Shanghai, China), progesterone (P4) (ml001945), and androgen (ml037278) ELISA kit were used to measure serum E2, P4, and androgen levels, respectively. Porcine E2 (ml002366) was used to measure the level of E2 secreted by porcine GCs. Briefly, 100 μL of HRP was added to the wells with 50 μL of standard and samples and incubated for 60 min at 37 ° C and 50 μL substrate A and B for 15 min. An OD value of 450 nm was obtained.

### 2.7. Lentivirus Delivery and TUNEL Assay

Three-week-old female C57BL/6J mice (*n* = 20) were randomly assigned into four groups: LV-NC, LV-lnc *SFFD*, sh-NC, and sh-lnc *SFFD*. Then, 1 × 10^7^ TU of synthesized lentiviral vectors (LV-lnc *SFFD* and sh-lnc *SFFD*) (Dongze Biotech, Guangzhou, China) was delivered once a week for 3 weeks according to our previous work [19].

The TUNEL assay was performed to detect the apoptosis of GCs in mouse ovaries using the TUNEL Apoptosis Assay Kit (Beyotime Biotech, Shanghai, China). Briefly, the ovaries in paraffin sections were treated with xylene, ethanol, protease K, and TUNEL reaction mixture, respectively.

### 2.8. Chromatin Immunoprecipitation (ChIP) and FISH Assay

ChIP was used to detect the H3K27ac level in the *SFFD* promoter region. According to the Pierce^TM^ ChIP kit (Thermo, Waltham, MA, USA), the chromatin fragments were incubated with H3K27ac antibody (Active Motif, Carlsbad, CA, USA) and IgG antibody (12-370, Millipore, Burlington, MA, USA), and the purified DNA was used for qPCR analyses; the primers are listed in Table 3.

The localization of *SFFD* in GCs was detected using lncRNA FISH probe mix and matching kits (RiboBio, Guangdong, China). The GCs were treated with 4% paraformaldehyde, 0.5% Triton X-100, the *SFFD* probe, and DAPI.

### 2.9. Dual-Luciferase Reporter Gene Assay

To assess the binding between *SFFD* and miR-202-3p, the *SFFD* 3′ untranslated region (UTR) segment (lnc *SFFD*-WT) and the mutation sites of the seed sequence (lnc *SFFD*-MUT) were established and inserted into the pmirGLO-reporter (Promega, Madison, WI, USA) with Sac1 and Sall endonuclease. Then, lnc *SFFD*-WT/MUT and mimics NC/miR-202-3p mimic were co-transfected into GCs for 48 h, respectively. The luciferase activity was performed using a luciferase detection kit (Promega, Madison, WI, USA).

To assess the target between *COX1* and miR-202-3p, the *COX1* 3′ UTR with binding sites of miR-202-3p, *COX1* 3′ UTR WT plasmid (*COX1* 3′ UTR-WT), and *COX1* 3′ UTR MUT plasmid (*COX1* 3′ UTR-MUT) were established, and *COX1*-WT/*COX1*-MUT plasmids and mimics NC/miR-202-3p mimic were co-transfected into GCs cultured in 24-well plates for 48 h, respectively.

### 2.10. Western Blot

The total protein was extracted from GCs or ovaries using RIPA lysis buffer (Beyotime, Shanghai, China) and quantified using a BCA protein assay kit (Beyotime, China). A total of 20 µg of sample protein after being electrophoresed on 4–20% SDS-PAGE gels was transferred to polyvinylidene difluoride (PVDF) membranes. Then, the PVDF membranes were sealed with 5% skimmed milk for 1 h at room temperature and incubated with the primary antibody at 4 °C overnight: HDAC1 (ab109411, abcam, 1:5000), HDAC2 (ab32117, abcam, 1:1000), HDAC3 (ab32369, abcam, 1:5000), PCNA (10205-2-AP, proteintech, 1:2000), CCND1 (26939-1-AP, proteintech, 1:1000), CASP3 (19677-1-AP, proteintech, 1:1000), HSD17B1 (25334-1-AP, proteintech, 1:2000), and Tubulin (11224-1-AP, proteintech, 1:5000). Secondary antibodies were incubated for 2 h at room temperature. ImageJ was used to measure the bands.

### 2.11. Amplification and Coding Ability of SFFD

The full-length amplification of *SFFD* was performed using the 5′/3′-RACE Kit, 2nd Generation (Roche, Basel, Switzerland). To determine the coding ability of *SFFD*, the GFP-WT, GFP-MUT, and Open Reading Frame of *SFFD* with GFP-MUT were inserted into the pcDNA3.1 vector. The primers and oligonucleotide sequences are listed in Appendix A.

### 2.12. Statistical Analysis

Data are exhibited as mean ± standard deviation (SD). Student’s *t*-test was used to measure differences. *p* < 0.05 or *p* < 0.01 were considered statistically significant.

## 3. Results

### 3.1. Hdac1 Is Required for Follicular Development and Sexual Maturity in Mice

To explore the role of histone acetylation on follicular development and sexual maturity, an HDAC inhibitor, TSA, was delivered into mice. Compared with DMSO (34.4 ± 1.19 d) and the blank group (34.5 ± 0.53 d), TSA significantly delayed the age of sexual maturity (40.5 ± 1.69 d, *p* < 0.01, *n* = 8) (Figure 1A,B). We found that TSA significantly raised the concentration levels of P4 (*p* < 0.05) and androgen (*p* < 0.01) but depressed the concentration level of E2 (*p* < 0.05), suggesting that the follicular development was blocked (Figure 1C). Moreover, the litter birth weight and litter sizes were significantly decreased at the first, second, and third parity by TSA (Figure 1D). Especially, compared to DMSO-treated mice, the ovaries of 45-day-old TSA-treated mice showed increased preantral follicles but bare corpora lutea (Figure 1E). The TUNEL assay further showed that TSA remarkably elevated the apoptosis of GCs (Figure 1F). Immunofluorescence (Figure 1G) and Western blot (Figure 1H) showed that TSA markedly decreased the protein level of Hdac1 (*p* < 0.01) but displayed an insignificant effect on the protein levels of Hdac2 and Hdac3.

### 3.2. HDAC1 Downregulation Promotes the Expression of CASP3 to Induce the Apoptosis of Porcine GCs

To understand the role of *HDAC1* in GCs, the porcine GCs were treated with TSA and *HDAC1*-siRNA. The results showed that the viability (Figure 2A,B) of GCs was significantly decreased by TSA in a dose-dependent manner at 24 h and 48 h. Although the mRNA level of *HDAC1* was significantly inhibited by 0.1, 0.25, and 0.5 μM TSA in GCs (Figure 2C), 0.1 μM TSA for 24 h was selected to reduce the toxic effects on GCs. We found that TSA-mediated *HDAC1* downregulation prominently reduced the mRNA levels of *PCNA* (*p* < 0.01), *CDK1* (*p* < 0.01), *CCNA1* (*p* < 0.05), and *CCND1* (*p* < 0.01) related to cell proliferation (Figure 2D) and remarkably elevated the mRNA levels of *CASP3* (*p* < 0.01), *CASP8* (*p* < 0.05), and *BAX* (*p* < 0.01) (Figure 2E). Moreover, TSA-mediated *HDAC1* downregulation markedly elevated the protein level of CASP3 (*p* < 0.05) (Figure 2F). Subsequently, EdU staining (Figure 2G) showed that TSA-mediated *HDAC1* downregulation significantly inhibited the proliferation of GCs. Flow cytometry (Figure 2H) and Caspase 3/7 activity assay (Figure 2I) further showed that TSA-mediated *HDAC1* downregulation significantly promoted the apoptosis and Caspase 3/7 activity of GCs. To further characterize the biological roles of *HDAC1* in GCs, three *HDAC1* small interfering RNA (*HDAC1*-siRNA1, *HDAC1*-siRNA2, and *HDAC1*-siRNA3) were constructed for *HDAC1* knockdown. Notably, *HDAC1*-siRNA2 markedly decreased the mRNA (*p* < 0.01) and protein (*p* < 0.05) levels of *HDAC1* (Figure 2J) and was used for *HDAC1* knockdown. We found that *HDAC1*-siRNA notably decreased the mRNA levels of *PCNA* (*p* < 0.05), *CDK1* (*p* < 0.05), and *CCND1* (*p* < 0.01) (Figure 2K) and prominently increased the mRNA levels of *CASP3* (*p* < 0.01) and *CASP8* (*p* < 0.05) (Figure 2L). Moreover, *HDAC1*-siRNA markedly elevated the protein level of CASP3 (*p* < 0.05) (Figure 2M). EdU staining (Figure 2N) showed that *HDAC1*-siRNA markedly arrested the proliferation of GCs. Flow cytometry (Figure 2O) and the Caspase 3/7 activity assay (Figure 2P) further showed that *HDAC1*-siRNA significantly promoted the apoptosis and Caspase 3/7 activity of GCs.

### 3.3. HDAC1-Mediated SFFD Targets COX1 by Sponging miR-202-3p

The induced apoptosis of GCs by *HDAC1* downregulation prompted us to investigate the underlying mechanisms. Specially, the transcriptomes of GCs treated with *HDAC1*-siRNA (named H) and siRNA-NC (named NH) were analyzed, and 43 differential expression lncRNAs were obtained (Figure 3A). Among these differential expression lncRNAs, MSTRG.23238.4, which we named as *SFFD*, was significantly downregulated in the *HDAC1*-siRNA group (Figure 3A). The qPCR further validated that TSA and *HDAC1*-siRNA significantly downregulated the expression of *SFFD* (Figure 3B). The 5′ and 3′ ends of *SFFD* were determined by RACE (Figure 3C), and the results showed that *SFFD* was 734 nt (NCBI accession number: OP120939) and located at chr MT (mitochondrion): 1548–2026. Subsequently, we found the pcDNA3.1-ORF with GFP-MUT did not show GFP fluorescence (Figure 3D), indicating that *SFFD* did not have protein-encoding potential. Additionally, *SFFD* was a cytoplasm-enriched lncRNA detected using FISH (Figure 3E) and a nuclear mass separation assay (Figure 3F). To clarify the *HDAC1*-mediated mechanism for the transcriptions of *SFFD*, the H3K27ac level of the *SFFD* promoter, including region 1 (−1718 to −1358 bp), region 2 (−1288 to −1031 bp), region 3 (−953 to −661 bp), region 4 (−639 to −282 bp), and region 5 (−279 to −37 bp), was detected in GCs treated with *HDAC1*-siRNA (Figure 3G). ChIP (Figure 3H) and ChIP-qPCR (Figure 3I) showed that *HDAC1*-siRNA significantly decreased the enrichment of H3K27ac in region 3, while also exhibiting insignificant effects on other regions.

It is predicted that *SFFD* may bind to miR-202-3p, miR-222, miR-7140-5p, miR-296-3p, and miR-199b-3p. The results of the RNA pull-down assay demonstrated that the interaction between miR-202-3p and *SFFD* exhibited the most favorable targeting relationship (Figure 3J). Moreover, the inhibitory effect of miR-202-3p on the transcription activity of *SFFD* was restored by mutating the binding site of miR-202-3p on *SFFD* (Figure 3K). PcDNA3.1-lnc *SFFD* (Figure 3L) and si-lnc *SFFD*-2 (Figure 3M) markedly increased and decreased the expression of *SFFD*, respectively. *SFFD* overexpression significantly inhibited the expression of miR-202-3p, while *SFFD* knockdown significantly promoted the expression of miR-202-3p (Figure 3N). Moreover, miR-202-3p mimic and the miR-202-3p inhibitor markedly increased and decreased the expression of miR-202-3p (Figure 3O), respectively. The inhibitory effect of miR-202-3p on the luciferase activity of *COX1* was restored via mutating the targeting sequence of miR-202-3p (Figure 3P). MiR-202-3p mimic significantly inhibited the mRNA level of *COX1*, while the miR-202-3p inhibitor significantly promoted the mRNA level of *COX1* (Figure 3Q).

### 3.4. SFFD Acts as ceRNA by Targeting miR-202-3p/COX1 to Control the Apoptosis of Porcine GCs

*HDAC1* downregulation induced the apoptosis of GCs, and *HDAC1* downregulation reduced the expression of *SFFD* to target *COX1* by sponging miR-202-3p. It was likely that the *SFFD*/miR-202-3p/*COX1* signaling axis regulated the apoptosis of GCs. Here, we found that *SFFD* overexpression notably increased the mRNA levels of *PCNA* (*p* < 0.01), *CDK1* (*p* < 0.01), *CCNA1* (*p* < 0.05), *CCND1* (*p* < 0.01) (Figure 4A), *CYP11A1* (*p* < 0.01), *HSD17B1* (*p* < 0.01), and *HSD3B1* (*p* < 0.01) (Figure 4B) but decreased the mRNA levels of *CASP3* (*p* < 0.01), *CASP8* (*p* < 0.05), and *CASP9* (*p* < 0.05) (Figure 4C), and miR-202-3p mimic partially attenuated these effects. However, *SFFD* overexpression remarkably decreased the protein level of CASP3 (*p* < 0.05) and did not show significant effects on the protein levels of CCND1 and HSD17B1 (Figure 4D). *SFFD* knockdown markedly decreased the mRNA levels of *PCNA* (*p* < 0.01), *CDK1* (*p* < 0.05), *CCND1* (*p* < 0.05) (Figure 4E), *CYP11A1* (*p* < 0.05), *HSD17B1* (*p* < 0.01), and *HSD3B1* (*p* < 0.01) (Figure 4F) but increased the mRNA levels of *CASP3* (*p* < 0.01), *CASP8* (*p* < 0.01), and *CASP9* (*p* < 0.05) (Figure 4G) and the protein level of CASP3 (*p* < 0.05) (Figure 4H), and the miR-202-3p inhibitor partially attenuated these effects. EdU staining further confirmed that *SFFD* overexpression remarkably promoted the proliferation of GCs, and miR-202-3p mimic partially attenuated the promoting effect (Figure 4I). Conversely, *SFFD* knockdown significantly inhibited the proliferation of GCs, and the miR-202-3p inhibitor partially attenuated the inhibitory effect (Figure 4J). The ELISA indicated that *SFFD* overexpression observably enhanced E2 secretion, while *SFFD* knockdown markedly inhibited E2 secretion, and miR-202-3p attenuated these effects (Figure 4K). Flow cytometry and the Caspase 3/7 activity assay showed that *SFFD* overexpression significantly inhibited the apoptosis (Figure 4L) and Caspase 3/7 activity (Figure 4M) of GCs, and miR-202-3p mimic partially attenuated the inhibitory effect. *SFFD* knockdown significantly promoted the apoptosis (Figure 4N) and Caspase 3/7 activity (Figure 4O) of GCs, and the miR-202-3p inhibitor partially attenuated this promoting effect.

### 3.5. MiR-202-3p Targeted COX1 to Regulate the Apoptosis of Porcine GCs

To verify whether *COX1* participates in the miR-202-3p-mediated pathway in GCs, three small interfering RNA (*COX1*-siRNA1, *COX1*-siRNA2, and *COX1*-siRNA3) were used for *COX1* knockdown. Among them, *COX1*-siRNA1 exhibited the most obvious inhibitory effect on the expression of *COX1* (Figure 5A). MiR-202-3p mimic remarkably decreased the mRNA levels of *PCNA* (*p* < 0.01), *CCND1* (*p* < 0.05), and *HSD17B1* (*p* < 0.05) (Figure 5B,C) and elevated the mRNA levels of *CASP3* (*p* < 0.05), *CASP8* (*p* < 0.05), and *CASP9* (*p* < 0.01) (Figure 5D). But the miR-202-3p inhibitor remarkably increased the mRNA levels of *PCNA* (*p* < 0.01), *CDK1* (*p* < 0.05), *CCNA1* (*p* < 0.05), *CCND1* (*p* < 0.01), *CYP11A1* (*p* < 0.01), *HSD17B1* (*p* < 0.01), and *HSD3B1* (*p* < 0.01) (Figure 5B,C), while it decreased the mRNA levels of *CASP3* (*p* < 0.01), *CASP8* (*p* < 0.05), and *CASP9* (*p* < 0.05) (Figure 5D), and *COX1*-siRNA partially attenuated the effect. EdU staining (Figure 5E) and ELISA (Figure 5F) further showed that miR-202-3p overexpression markedly arrested the proliferation and E2 secretion of GCs. Conversely, miR-202-3p knockdown exhibited the opposite effect, and *COX1*-siRNA partially attenuated the effect. Flow cytometry (Figure 5G) exhibited that miR-202-3p knockdown remarkably inhibited the apoptosis of GCs, and *COX1*-siRNA partially weakened this inhibitory effect.

### 3.6. SFFD as ceRNA for miR-202-3p-Cox1 Accelerates Sexual Maturity in Mice

*SFFD* was found to exist in mouse ovaries detected by FISH (Figure 6A). We found that the mRNA levels of *HDAC1* and *SFFD* were significantly elevated in porcine GCs treated with GDF9 + BMP15 (Figure 6B,C), suggesting that the transcription of *SFFD* epigenetically mediated by *HDAC1* was highly implicated in follicular development and sexual maturity. LV-lnc *SFFD* and sh-lnc *SFFD* were delivered to determine the effect of *SFFD* on the development of follicles and sexual maturity in mice. Especially, LV-lnc *SFFD* was found to significantly promote the expressions of *SFFD* and *Cox1* but reduce the expression of miR-202-3p (Figure 6D). Nevertheless, sh-lnc *SFFD* markedly inhibited the expressions of *SFFD* and *Cox1* and remarkably elevated the expression of miR-202-3p (Figure 6D). LV-lnc *SFFD* significantly promoted the mRNA levels of *Ccnd1* (*p* < 0.01) and *Hsd17b1* (*p* < 0.05) and inhibited the mRNA level of *Casp3* (*p* < 0.05) (Figure 6E); it also elevated the protein level of Hsd17b1 (*p* < 0.05) (Figure 6F). Conversely, sh-lnc *SFFD* markedly reduced the mRNA levels of *Pcna* (*p* < 0.01), *Cdk1* (*p* < 0.05), and *Ccnd1* (*p* < 0.05) and promoted the mRNA levels of *Casp3* (*p* < 0.01), *Casp8* (*p* < 0.05), and *Casp9* (*p* < 0.05) (Figure 6E). Sh-lnc *SFFD* significantly decreased the protein level of Ccnd1 (*p* < 0.05) and increased the protein level of Casp3 (*p* < 0.05) (Figure 6F). Moreover, compared with sh-NC (47.6 ± 2.61 d vs. 40.2 ± 1.48 d, *p* < 0.01, *n* = 5), sh-lnc *SFFD* markedly delayed sexual maturity, while LV-lnc *SFFD* did not show a significant effect on sexual maturity compared with LV-NC (37.8 ± 1.64 d, vs. 39.4 ± 1.14 d, *p* < 0.01, *n* = 5) (Figure 6G). The E2 concentration in the LV-lnc *SFFD* group was markedly increased, while that in the sh-ln2c *SFFD* group was remarkably decreased (Figure 6H). Notably, the number of preantral follicles in the sh-lnc *SFFD* group was prominently increased, but the number of corpora lutea was markedly reduced (Figure 6I). TUNEL further determined that sh-lnc *SFFD* remarkably enhanced the apoptosis of GCs (Figure 6J).

## 4. Discussion

At present, the epigenetic regulation mechanism of sexual maturity mainly focuses on DNA methylation. Lomniczi et al. found that the inhibition of DNA methylation resulted in rats failing to ovulate and a delay in sexual maturity [14]. However, the mechanism by which histone acetylation regulates sexual maturity remains to be further clarified. To explore the role of histone acetylation on follicular development and sexual maturity in mice, TSA, an effective histone deacetylating inhibitor, was delivered into mice. In the present study, we found that TSA remarkably delayed the age of sexual maturity (Figure 1B) and fertility (Figure 1D) in mice. Consistent with our study, gilts that reach sexual maturity at a younger age exhibit higher fertility in their lifetime than gilts attaining sexual maturity at an older age [28]. Furthermore, the ovaries of TSA-treated mice showed increased preantral follicles but fewer corpora lutea, indicating that these TSA-treated mice did not have mature follicles and had not ovulated, and they failed to reach sexual maturity (Figure 1E). The apoptosis of follicular GCs in TSA-treated mice was remarkably higher than DMSO-treated mice (Figure 1F). In addition, TSA significantly decreased the protein level of Hdac1 in mouse ovary, but Hdac2 and Hdac3 did not show significant changes (Figure 1G,H). To further verify the effect of histone acetylation on the function of GCs, we treated porcine GCs with TSA and *HDAC1*-siRNA, respectively. Both TSA-mediated *HDAC1* knockdown and *HDAC1*-siRNA prominently arrested the proliferation of GCs (Figure 2G,N) and enhanced the apoptosis of GCs (Figure 2H,O). Consistent with previous studies, the HDAC inhibitor induces apoptosis and cell death [29], and the knockdown of *HDAC1* reduces proliferation and enhances apoptosis in various cell types [30]. It is concluded that *HDAC1* is involved in sexual maturity by mediating the apoptosis of GCs.

LncRNAs are generally considered to be transcripts that have over 200 nucleotides with no protein coding ability [31]. It has been demonstrated that lncRNAs sponge miRNAs, a category of transcripts termed competing endogenous RNAs, through base–pairing interactions [32] and mediate the growth and function of GCs to participate in follicular development [33,34] and sexual maturity [35,36]. For example, lncRNA FDNCR enhances the apoptosis of GCs through the miR-543-3p/DCN/TGF-β axis in sheep [37], and lncRNA Gm2044 enhances E2 synthesis through targeting miR-138-5p in mouse pre-antral GCs [38]. The downregulation of lncRNA ZNF674-AS1 markedly reduces the proliferation of GCs and follicular growth [39]. In the present study, the knockdown of *HDAC1* remarkably decreased the expression of *SFFD* (Figure 3A,B). Epigenetic modifications have been reported to regulate the expression of lncRNA by affecting promoter utilization [40,41]. Previous studies have indicated that *HDAC1* epigenetically regulates the H3K27ac level at the gene’s promoter [42,43]. To explore the mechanism by which *HDAC1* inhibited the expression of *SFFD*, we analyzed the histone acetylation of the *SFFD* promoter and found that *HDAC1*-siRNA significantly decreased the H3K27ac level in the −953/−661 region at the *SFFD* promoter (Figure 3I). It has been shown that *HDAC1* exhibits deacetylase activity to inhibit transcription [44]. H3K27ac, a marker of transcriptional activation, is reported to be regulated by *HDAC1* [45] and participates in follicular maturation and ovulation [46]. Consistent with our results, in BrCa cells, *HDAC1* knockdown inhibits the expressions of super-enhancers associated with oncogenes through decreasing the H3K27ac level [45]. Therefore, we speculate that the mechanism by which *HDAC1* knockdown leads to reduced levels of H3K27ac in the *SFFD* promoter may be the presence of a super-enhancer in the *SFFD* promoter region. But the mechanism of this needs to be further explored. The level of H3K27ac is significantly induced during follicular maturation in mice [46]. LncRNA *ROCKI* reduces the expression of *MARCKS* through facilitating the *HDAC1*-mediated removal of H3K27ac [42]. These observations suggest that H3K27ac targets the −953/−661 region to epigenetically regulate the transcription of *SFFD*.

In the present study, *SFFD* was found to promote the proliferation (Figure 4I,J) and E2 secretion (Figure 4K) of GCs by regulating the expression of *CCND1* and *HSD17B1*, respectively, and regulates the expression of *CASP3* to inhibit the apoptosis of GCs (Figure 4L,N). The molecular function of lncRNA mainly depends on its subcellular localization [47]. To clarify the downstream mechanism of *SFFD*, the localization of *SFFD* in GCs was explored. The results showed that *SFFD* existed in both the cytoplasm and nucleus, but it was more abundant in the cytoplasm (Figure 3E,F). Therefore, we hypothesized that *SFFD* may exert its cellular functions by acting as a sponge for miRNA. Through bioinformatics tools, miR-202-3p was predicted to be one of the potential targets of *SFFD*, and the target relationship between them was further verified using RNA pull-down (Figure 3J) and dual luciferase analysis (Figure 3K). MiR-202-3p, as a member of the let-7 family, is highly conserved across animal species and has been reported to increase cell apoptosis while decreasing cell proliferation [48,49]. MiR-202-3p enhances apoptosis and arrests proliferation via binding *CCND1* in human Sertoli cells [48]. An elevated level of miR-202-3p has been observed in atretic follicles compared to chicken yellowish follicles, indicating its potential involvement in follicular atresia [50]. In mouse spermatogonial stem cells, miR-202-3p has been shown to arrest the cell cycle [51]. Consistent with previous studies, we revealed that miR-202-3p reduced the proliferation (Figure 5E) and E2 secretion (Figure 5F) and induced the apoptosis of GCs (Figure 5G). Furthermore, miR-202-3p partially reversed the cellular function mediated by *SFFD*. MiRNAs, as a vital epigenetic regulator, are evolutionarily conserved and exert an inhibitory effect on the expression of mRNA via a base–pairing interaction with the 3′ UTRs of target mRNAs [52,53]. In the present study, the interaction between *COX1* and miR-202-3p was verified through dual luciferase analysis (Figure 3P). *COX1*, one of the rate-limiting enzymes of prostaglandin synthesis, exhibits a vital role in cellular proliferation [54], apoptosis [55], and follicular angiogenesis [56] and development [57]. In this study, we found that *COX1*-siRNA partially attenuated the pro-proliferation and anti-apoptotic effects of the miR-202-3p inhibitor (Figure 5E,G). GDF9 and BMP15, the oocyte-secreting factors, had been identified to be indispensable for normal follicular development and sexual maturity [46]. In our study, GDF9 + BMP15 was found to markedly promote the mRNA levels of *HDAC1* and *SFFD* in porcine GCs (Figure 6B,C), which also provides further evidence that *HDAC1*-mediated *SFFD* is involved in follicular development and sexual maturation. To characterize the in vivo function of *SFFD*, *SFFD* overexpression and knockdown were simulated by the direct injection of lentiviral in mouse ovary. We found that *SFFD* knockdown notably blocked the follicular maturation (Figure 6I), significantly inhibited E2 secretion (Figure 6H), and markedly delayed the age of sexual maturity in mice (Figure 6G). These findings provide valuable insights into the therapeutic potential of *SFFD* in ovarian follicles and sexual maturation.

## 5. Conclusions

In conclusion, our findings demonstrate that *HDAC1* knockdown markedly reduces the H3K27ac level of the −953/−661 region to epigenetically inhibit the transcription of *SFFD*. *SFFD* functions by targeting miR-202-3p to elevate the expression of *COX1*, enhances proliferation and E2 secretion, and inhibits the apoptosis of GCs to regulate follicular development and sexual maturity (Figure 7). Collectively, our results suggest that *SFFD* mediated by *HDAC1* may serve as a potential molecular therapeutic approach for sexual maturity.

## Figures and Tables

**Figure 1 cells-12-02734-f001:**
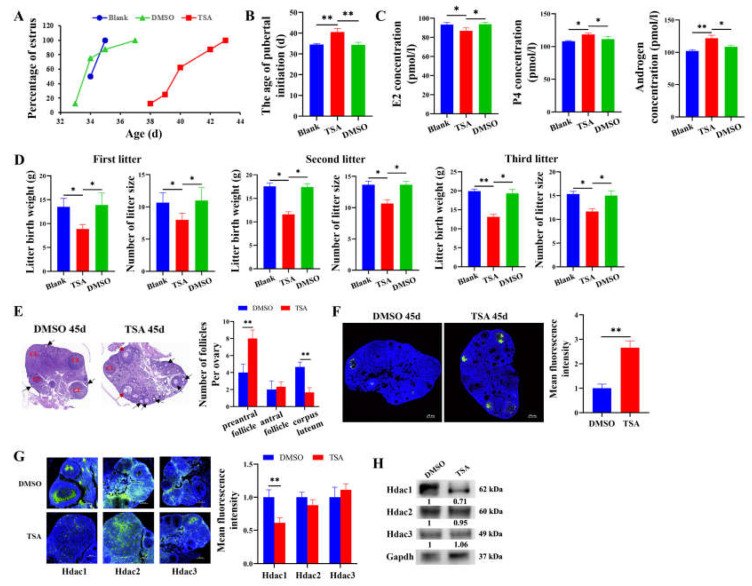
Downregulation of Hdac1 blocks follicular development and sexual maturity in mice. (**A**) The percentage of mouse estrus was assessed in the blank, DMSO, and TSA group (*n* = 8). (**B**) The age of mouse puberty was counted in the blank, DMSO, and TSA group (*n* = 8). (**C**) Effects of TSA on the concentrations of E2, P4, and androgen in the serum of mice. (**D**) Effects of TSA on the litter birth weight and number of litter size in the 1st to 3rd parities of mice. (**E**–**H**) Effects of TSA on the follicular development, apoptosis of follicular GCs, and levels of Hdac1, Hdac2, and Hdac3 in mice. Black arrows represent preantral follicles, red arrows represent antral follicles, and CL represents corpus luteum. * represents *p* < 0.05, ** represents *p* < 0.01.

**Figure 2 cells-12-02734-f002:**
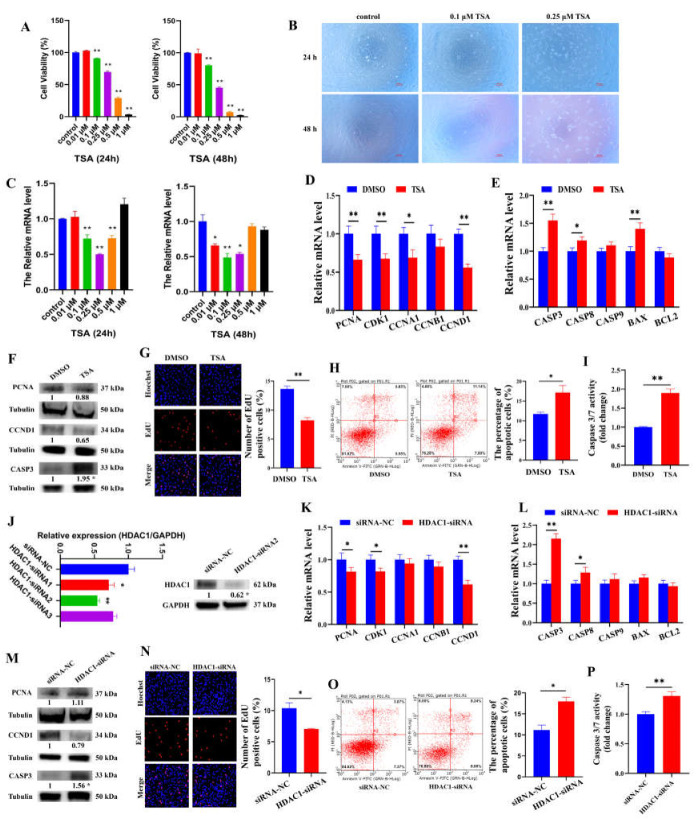
*HDAC1* regulates the proliferation and apoptosis of GCs. (**A**) Effects TSA treatment at concentration of 0.01, 0.1, 0.25, 0.5, and 1 μM for 24 and 48 h on the viability of porcine GCs. (**B**) Micrographs of GCs treated with 0.1 and 0.25 μM TSA at 24 and 48 h. (**C**) Effects of 0.01, 0.1, 0.25, 0.5, and 1 μM TSA treatment on the expression of *HDAC1* for 24 and 48 h. (**D**–**F**) The mRNA and protein levels of cell cycle and cell apoptosis-related genes in GCs treated with TSA. (**G**,**H**) The proliferation and apoptosis of GCs treated with TSA were assessed using EdU and flow cytometry. (**I**) Effect of TSA on the activity of Caspase 3/7 in porcine GCs was detected using Caspase 3/7 activity assay. (**J**) The interference efficiency of *HDAC1*. (**K**–**M**) The mRNA and protein levels of cell cycle and cell apoptosis-related genes in GCs treated with *HDAC1*-siRNA. (**N**,**O**) The proliferation and apoptosis of GCs treated with *HDAC1*-siRNA were assessed using EdU and flow cytometry. (**P**) Effect of *HDAC1*-siRNA on the activity of Caspase 3/7 in porcine GCs was detected using Caspase 3/7 activity assay. * indicates *p* < 0.05, ** indicates *p* < 0.01.

**Figure 3 cells-12-02734-f003:**
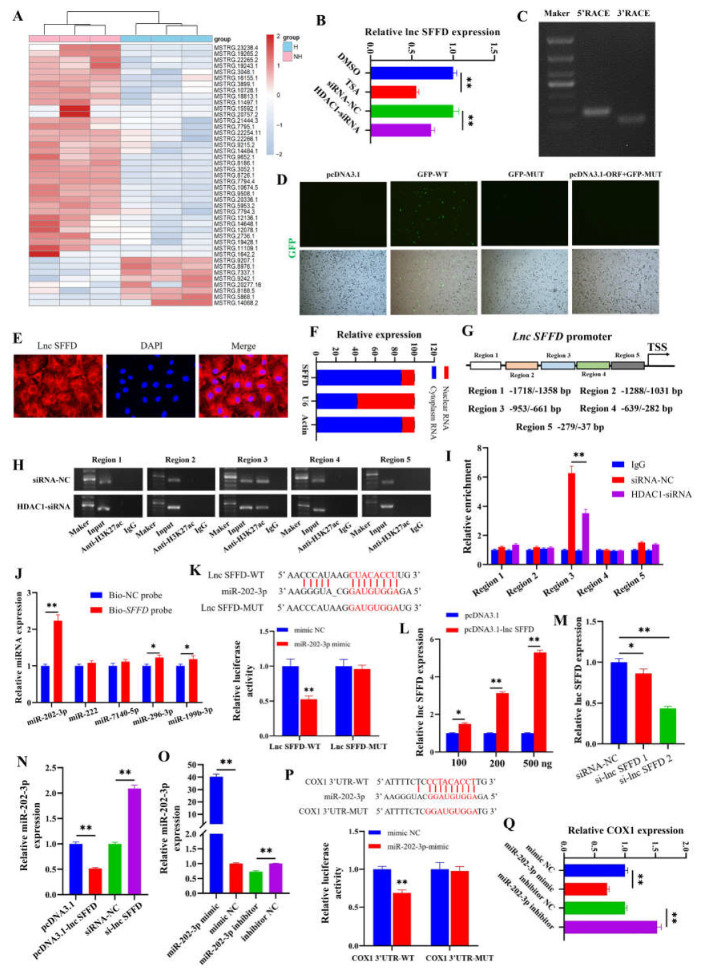
*HDAC1*-mediated *SFFD* targets *COX1* by sponging miR-202-3p. (**A**) Heat maps of differentially expressed lncRNAs between *HDAC1*-siRNA and siRNA-NC. (**B**) Effect of TSA or *HDAC1*-siRNA on the expression of *SFFD* in GCs. (**C**) 5′ RACE and 3′ RACE of *SFFD*. (**D**) GFP fluorescence of GCs transfected with plasmids for 48 h. (**E**,**F**) FISH and qPCR were performed to exhibit the localization of *SFFD* in GCs. (**G**) Schematic distribution of *SFFD* promoter. (**H**,**I**) The enrichment of H3K27ac on *SFFD* promoter in GCs treated with *HDAC1* knockdown was assessed by ChIP-PCR and ChIP-qPCR. (**J**) The expression of miR-202-3p, miR-222, miR-7140-5p, miR-296-3p, and miR-199b-3p in the *SFFD* pull-down complex. (**K**) Fluorescence activity of GCs co-transfected with *SFFD*–WT/MUT and miR-202-3p mimic/mimic NC. (**L**–**N**) Effect of pcDNA3.1-lnc *SFFD* and si-lnc *SFFD* on the expression of *SFFD* and miR-202-3p in GCs. (**O**) Effect of miR-202-3p mimic and inhibitor on the expression of miR-202-3p. (**P**) Fluorescence activity of GCs co-transfected with *COX1* 3′UTR–WT/MUT and miR-202-3p mimic/mimic NC. (**Q**) Effect of miR-202-3p mimic and inhibitor on the expression of *COX1* in GCs. * indicates *p* < 0.05, ** indicates *p* < 0.01.

**Figure 4 cells-12-02734-f004:**
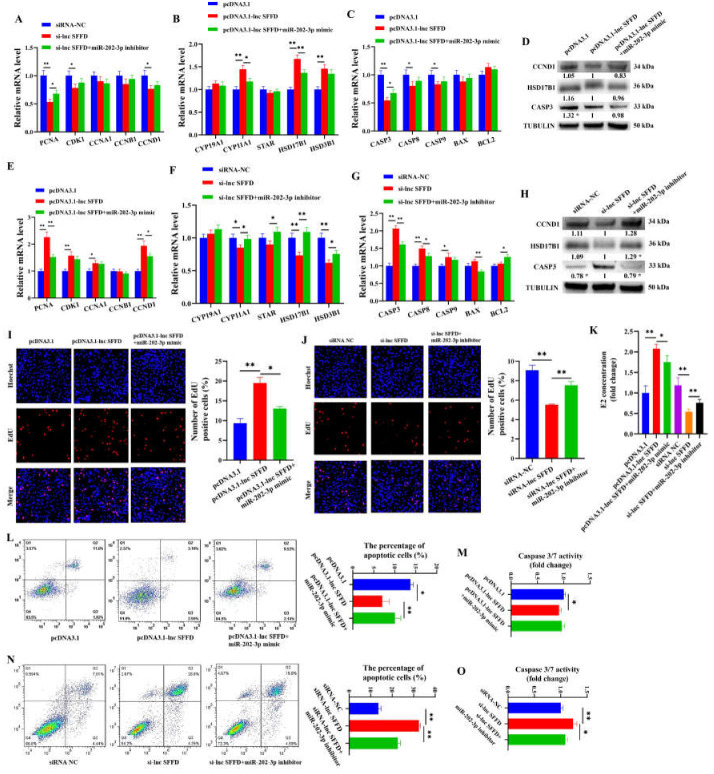
*SFFD* as ceRNA for miR-202-3p-*COX1* regulates the proliferation and apoptosis of porcine GCs. (**A**–**D**) Effects of *SFFD* overexpression and miR-202-3p mimic on the mRNA and protein levels of genes related to cell cycle, E2 secretion, and apoptosis. (**E**–**H**) Effects of *SFFD* knockdown and miR-202-3p inhibitor on the mRNA and protein levels of genes related to cell cycle, E2 secretion, and apoptosis. (**I**) Effects of *SFFD* overexpression and miR-202-3p mimic on the proliferation of GCs. (**J**) Effects of *SFFD* knockdown and miR-202-3p inhibitor on the proliferation of GCs. (**K**) Effects of *SFFD* overexpression with miR-202-3p mimic and *SFFD* knockdown with miR-202-3p inhibitor on the E2 secretion of GCs. (**L**,**M**) Effects of *SFFD* overexpression and miR-202-3p mimic on the apoptosis and Caspase 3/7 activity of GCs. (**N**,**O**) Effects of *SFFD* knockdown and miR-202-3p inhibitor on the apoptosis and Caspase 3/7 activity of GCs. * indicates *p* < 0.05, ** indicates *p* < 0.01.

**Figure 5 cells-12-02734-f005:**
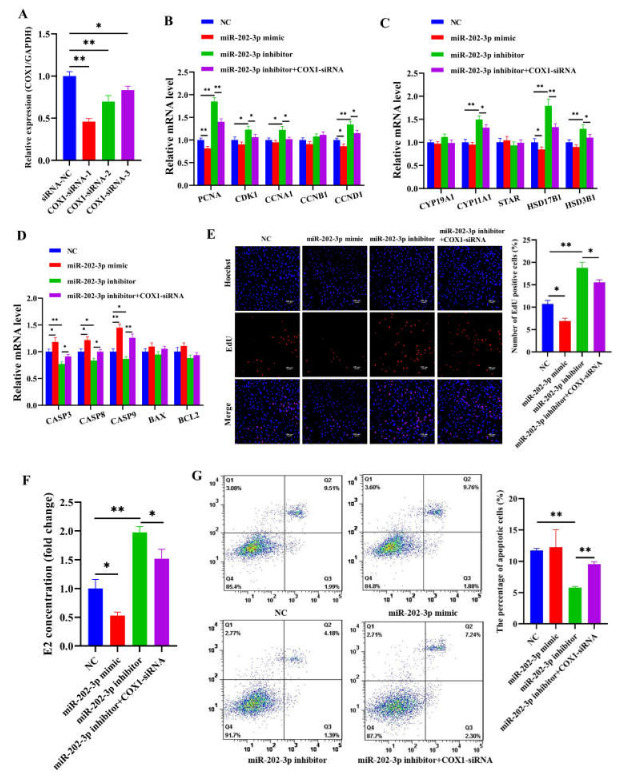
MiR-202-3p targeted *COX1* to regulate the apoptosis of porcine GCs. (**A**) Interference efficiency of *COX1*-siRNA. (**B**–**D**) Effects of miR-202-3p and *COX1*-siRNA on the mRNA levels of cell proliferation-related genes, E2 secretion-related genes, and apoptosis-related genes. (**E**–**G**) Effects of miR-202-3p and *COX1*-siRNA on the proliferation, E2 secretion, and apoptosis of GCs. * indicates *p* < 0.05, ** indicates *p* < 0.01.

**Figure 6 cells-12-02734-f006:**
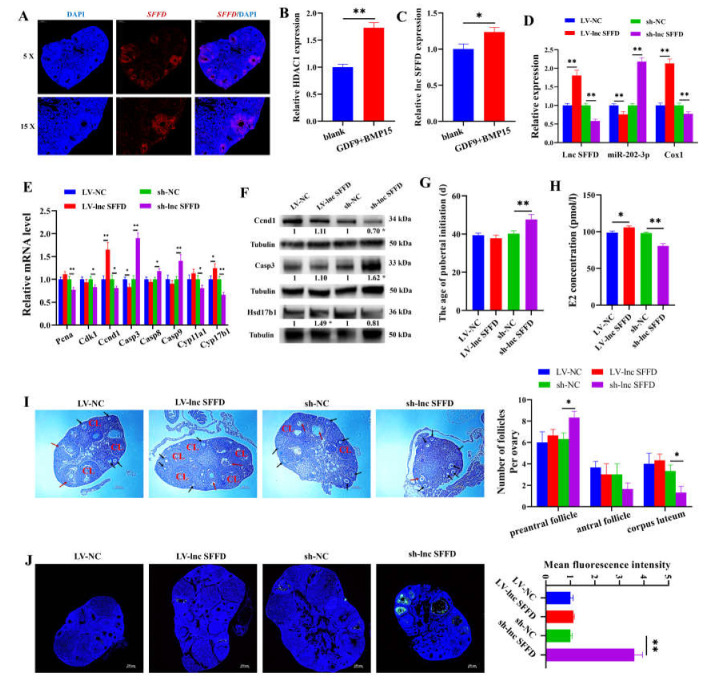
*SFFD* as ceRNA for miR-202-3p-*Cox1* accelerates sexual maturity in mice. (**A**) The expression of *SFFD* in mouse ovary detected by FISH. (**B**,**C**) Effects of GDF9 + BMP15 on the mRNA levels of *Hdac1* and *SFFD*. (**D**–**F**) The expressions of *SFFD*, miR-202-3p, *Cox1*, *Pcna*, *Cdk1*, *Ccnd1*, *Casp3*, *Casp8*, *Casp9*, *Cyp11a1*, and *Hsd17b1* in mouse ovary with *SFFD* overexpression and inhibition. (**G**–**J**) Effects of LV-lnc *SFFD* and sh-lnc *SFFD* on the percentage of mice that had a vaginal opening in each group (*n* = 5), E2 concentration, HE staining, and the apoptosis of follicular GCs. Black arrows represent preantral follicles, red arrows represent antral follicles, and CL represents corpus luteum. * indicates *p* < 0.05, ** indicates *p* < 0.01.

**Figure 7 cells-12-02734-f007:**
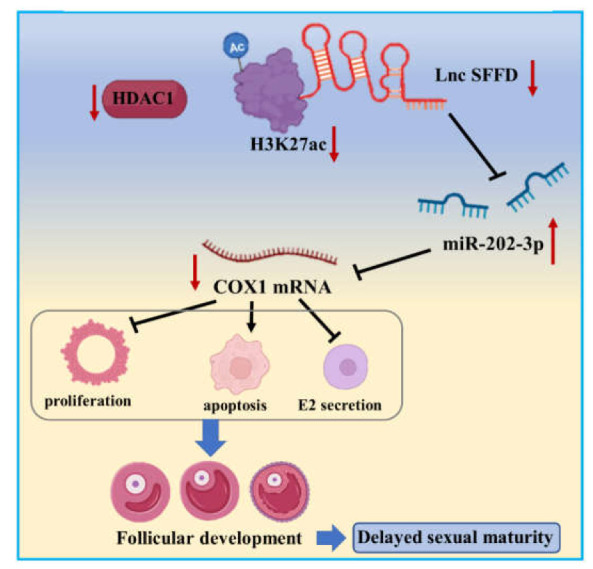
The mechanistic scheme of *HDAC1*-mediated H3K27ac level of *SFFD* to impede follicular development and sexual maturity.

**Table 1 cells-12-02734-t001:** Primers used for qPCR in pig.

Gene Name	Primer Sequences (5′ to 3′)	Size (bp)	Accession Number
HDAC1	F: AAGATGGCGCAGACTCAGGG	122	XM_013999116.2
	R: TTGTGAGTCATGCGGATTCGG		
HDAC2	F: ACTGGGCTGGAGGATTACAC	250	XM_001925318.6
	R: CCTGCACCAATATCCCTCAA		
HDAC3	F: ATTTCTACGACCCCGACGTG	105	NM_001243827.1
	R: AAAGACCGTAGTGCAGGACC		
PCNA	F: TCGTTGTGATTCCACCACCAT	278	NM_001291925.1
	R: TGTCTTCATTGCCAGCACATTT		
CDK1	F: AGGTCAAGTGGTAGCCATGAA	225	NM_001159304.2
	R: TCCATGAACTGACCAGGAGG		
CCNA1	F: GCGCCAAGGCTGGAATCTAT	196	XM_005668339.3
	R: CCTCAGTCTCCACAGGCTAC		
CCNB1	F: ACGGCTGTTAGCTAGTGGTG	236	NM_001170768.1
	R: GAGCAGTTCTTGGCCTCAGT		
CCND1	F: CTTCCATGCGGAAGATCGTG	234	XM_021082686.1
	R: TGGAGTTGTCGGTGTAGATGC		
CASP3	F: ACATGGAAGCAAATCAATGGAC	154	NM_214131.1
	R: TGCAGCATCCACATCTGTACC		
CASP8	F: GAGCCTGGACTACATCCCAC	283	NM_001031779.2
	R: GTCCTTCAATTCCGACCTGG		
CASP9	F: GCTGAACCGTGAGCTTTTCA	161	XM_003127618.4
	R: CCTGGCCTGTGTCCTCTAAG		
BAX	F: ACTTCCTTCGAGATCGGCTG	184	XM_013998624.2
	R: AAAGACACAGTCCAAGGCGG		
BCL2	F: GATGCCTTTGTGGAGCTGTATG	145	XM_021099593.1
	R: CCCGTGGACTTCACTTATGG		
STAR	F: CGACGTTTAAGCTGTGTGCT	136	NM_213755.2
	R: ATCCATGACCCTGAGGTTGGA		
CYP11A1	F: CTAAAACCCCTCGCCCCTTC	199	NM_214427.1
	R: GCCACATCTTCAGGGTCGAT		
CYP19A1	F: CTGAAGTTGTGCCTTTTGCCA	139	NM_214429.1
	R: CTGAGGTAGGAAATTAGGGGC		
HSD3B1	F: ATCTGCAGGAGATCCGGGTA	216	NM_001004049.2
	R: CCTTCATGACGGTCTCTCGC		
HSD17B1	F: GTCTGGCATCTGACCCATCTC	166	NM_001128472.1
	R: CGGGCATCCGCTATTGAATC		
Lnc SFFD	F: GCTGAATTGGCAAGGGTTGG	201	
	R: AGCTAGGACCCAAACTGGGA		
COX1	F: ACGGCACACGACTACATCAG	122	XM_001926129.6
	R: GGCAACTGCTTCTTCCCTTTG		

**Table 2 cells-12-02734-t002:** Primers used for qPCR in mouse.

Gene Name	Primer Sequences (5′ to 3′)	Size (bp)	Accession Number
Cox1	F: AGAGGTGACAACTGGAGGGAG	253	NM_008969.4
	R: CAACAGGGATTGACTGGTGAG		
Casp3	F: TGGCGTGTGCGAGATGAG	211	NM_009810.3
	R: TTGTTGTTCTCCATGGTCAC		
Casp9	F: GGGAAGCCCAAGCTCTTCTT	234	NM_015733.5
	R: CCAGGAGACAAAACCTGGGAA		
Casp8	F: CGGGAAAAGGGGATGTTGGA	202	NM_009812.2
	R: CCAACTCGCTCACTTCTTCTGA		
Pcna	F: GTGAACCTGCAGAGCATGGA	216	NM_011045.2
	R: TGGTGCTTCGAATACTAGTGC		
Cdk1	F: TGGGGTGTTGTTTCCACAGTT	268	NM_007659.4
	R: AGGGGCTGAGACCAATGGAG		
Ccnd1	F: GCCATCCATGCGGAAAATCG	205	NM_001379248.1
	R: GGCAGTCAAGGGAATGGTCT		
Cyp11a1	F: TACTAACCTAGCCCGCCTCG	235	NM_001346787.1
	R: GAGTCCCATGCTGAGCCAGA		
Hsd17b1	F: AGATTGCCAGCAGACACAACA	273	XM_006532297.3
	R: CAACAATGGTCCCTGTGCCTT		

**Table 3 cells-12-02734-t003:** Primers used for ChIP-PCR.

Primer Name	Primer Sequences (5′ to 3′)
Region 1	F1: GTGAGGTTGGTCCCATTTC
	R1: GGTGACCTCGGAGTACAA
Region 2	F2: GAATAGTTAATTAAAGCTCC
	R2: CACCTCTAGCATTACTAGT
Region 3	F3: CCTGTGTTGGGTTAACAATG
	R3: ATACTACCATAGTAGGCCT
Region 4	F4: GTCAAGGTTGTATCCGTT
	R4: GAATAAAATTCAAAGTAAG
Region 5	F5: CATCTATCCCTTACGGTACTA
	R5: ACTGGAAAGTGTGCTTGGA

## Data Availability

The data are available from the corresponding author on reasonable request.

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
