# Peer review of "HDAC1-Mediated lncRNA Stimulatory Factor of Follicular Development to Inhibit the Apoptosis of Granulosa Cells and Regulate Sexual Maturity through miR-202-3p-COX1 Axis"

_cells, 2023, doi:10.3390/cells12232734_

Round 1

Reviewer 1 Report (Previous Reviewer 1)

Comments and Suggestions for Authors

My comments were taken into account by the authors and I believe that now the article is written more clearly and accurately, with all the necessary reservations.

Reviewer 2 Report (Previous Reviewer 3)

Comments and Suggestions for Authors

The authors applied the proposed changes, thanks to which the manuscript has improved readability and increased its scientific value. I accept all changes and do not report any further changes. Therefore, I recommend this manuscript for publication in the Cells.

This manuscript is a resubmission of an earlier submission. The following is a list of the peer review reports and author responses from that submission.

Round 1

Reviewer 1 Report

Comments and Suggestions for Authors

This is a very interesting study, of great interest both from the point of view of the discovery of fundamental mechanisms of epigenetic regulation of follicle maturation, and from the point of view of the practical application of the research results. However, before the work can be appropriate for publication in Cells, the authors should clarify a number of points both in the design of the study and in the interpretation of the results.

It is not at all clear from the introduction and experimental design why some of the experiments were performed on porcine GCs. In particular, porcine GCs were used to elucidate the role of histone acetylation as a result of treatment with either TSA or HDAC1-siRNA.

Moreover, in the discussion (lines 370-371) authors are talking about rats! I believe this is a typo, but may nonetheless be misleading to the reader.

The results of Western blotting on the histograms in the article look good, however, in the original images, repetition 2 almost everywhere shows the opposite results, starting from the very first - when treated with TSA, the HDAC1 content is significantly higher than when treated with DMSO, and so on for almost all proteins in this repetition.

The big question is raised by the discussion paragraph on lines 390 – 401. The paper [43] showed that HDAC1 inhibition leads to a global increase in H3K27ac, with the exception of super-enhancers in the promoter regions of stem cell associated transcription (SCTF) genes. The authors cite this work as an explanation for the decrease in H3K27ac in region 3 of the SFFD promoter region. However, the study does not present any data that would allow us to consider SFFD as an SCTF gene and, moreover, there is no data on super-enhancers in its promoter region. Authors should provide these data.

Reviewer 2 Report

Comments and Suggestions for Authors

LncRNA SFFD mediated by HDAC1 inhibits the apoptosis of 2 granulosa clls to control sexual maturity via targeting COX1 by 3 sponging miR-202-3p

By Zhou et al.

The use of the English language needs to be improved considerably. As it stands now, the quality is below the level of acceptation. Indeed this is already visible in the title which is confusing (…mediated by…via…by sponging.) and contains a typo (clls instead of cells).

In general, the author are focused on ‘sexual maturity’. This seems very much dependent on hormone levels (E2), assuming that also later in life the female (mice) suffer from reduced fertility (fewer ovulations). This would have a higher impact, in terms of fertility, than onset of sexual maturity, starting a few days later. What is the information on levels of fertility of postpubertal (female) animals? It appears to me that SFFD and HDAC1 are important for correct androgen (estrogen in particular) levels throughout life. Or is only functional at the onset of puberty?

The figures are too small. Text in the figures, such as on graph axes, can not/hardly be read.

Genes should be in italics, mouse genes starting with a capital, rest in lower case. Porcine genes completely in capitals. Protein names should not be in italics.

Abbreviations should be explained the first time they are used (for instance GC).

Throughout the manuscript the word ‘obviously’ is incorrectly used. Possibly the authors mean ‘significantly’ or ‘remarkably’ or  ‘interestingly’.

For figure 2, data from porcine granulosa cells were used.  The origin of the cells, and the method of cell culture also of these cells should be properly explained.

2.2 For quantitative PCR, the M&M should describe the method of reverse transcription, and cycling conditions (e.g. number of cycles, temperatures).

2.10 It should be described how the antibodies bound to the membrane were detected.

3.1 Compared with DMSO (34.4±1.19 d)… What is the unit here? How is sexual maturity defined?

Figure 1A, percentage of estrus. N=8. It seems not correct that at one day (e.g. 34, blank group) there are 4 dots. On this day, there is only one percentage. So day 34, blank group would be 50% (4 from 8).

Figure 1E: are these total numbers of follicles per ovary? That is hard to accept, the numbers of preantral follicles will be much higher. Or is this number follicles counted in one ovary section? Same for Fig 6I

Figure 2A: the bar graphs and line graph essentially show the same data. Choose either bard graph or the line graph, but not both.

Figure 6B, C: are these data from whole ovaries, or from cultured granulosa cells?

Figure 6G How was vaginal opening determined? Double blinded? In the Methods we can read that ‘8 mice in each group were used to determine age of pubertal initiation and their ovaries were used for HE staining’ How were these 8 mice selected? Randomly?

Estrus levels in Fig 6H: are these levels in blood, follicular fluid? How were levels determined, this information is lacking in the methods section.

In the Discussion part it is stated that SFFD was more abundant in the cytoplasm. In figure 7 however, it is depicted in the nucleus. This is confusing.

Minor

2.4 EdU assay:  10 um EDU should be 10 µM EdU

Comments on the Quality of English Language

It is advised to have the manuscript corrected by someone fluent in the English language. As it stands now, it is below the level of acceptance

Reviewer 3 Report

Comments and Suggestions for Authors

In the presented publication, the authors test the hypothesis that histone acetylation might serve as a potential therapeutic target to control sexual maturity. For their research, they used a mouse model. They characterized the mechanisms for histone acetylation controlling sexual maturity with Trichostatin A (TSA), an HDAC inhibitor. They found that SFFD knockdown appeared to be responsible for the delayed sexual maturity. They also try to show that the results presented provide novel evidence for HDAC1-mediated-lncRNAs in the treatment of delayed sexual maturity. The authors present interesting research that applies to human medicine and deals with problems of reaching sexual maturity. Unfortunately, the authors did not avoid many errors in the construction of the publication that reduced the readability of the publication and its scientific value. All the objections are listed below.

Lines 63-71: The authors have not quite clearly presented all the aims they undertook to investigate in this publication. Part of this paragraph is a summary of the results obtained rather than a presentation of the aims.

Line 74: What is the purpose of using 4 male mice?

Table 1. The authors in materials and methods don't mention anything about qPCR studies on a pig model.

Lines 95-99: When presenting each method, it would be appropriate to indicate in one sentence for what purpose the method was used, for example, in point 2.5.

Lines 106-110: What did the authors determine by the ELISA method using the indicated kits?

Lines 111-118: In section 2.5, the authors described the use of flow cytometry to determine apoptosis. The TUNEL method is also used to test for apoptosis. Why in this case did the authors use another method to test apoptosis unless the method was used for another purpose?

All the graphs and images shown are too small and unreadable.

Presenting the results, the authors indicate that some factors decreased or increased the expression of some genes or the concentration of some factors, but they do not provide values of P<? However, the graphs, indicate differences in the number of asterisks. This introduces confusion in the description of the results.

In the "Results" chapter, the authors should present only the results obtained in the experiments. However, in this publication, this chapter also includes elements that should be included in the "Discussion" chapter such as conclusions and citations in lines: 165-167. I suggest checking out the entire chapter in this regard.